# Quantitative bioluminescence assay for measuring *Bacillus cereus* nonhemolytic enterotoxin complex

**Reuven Rasooly**[ORCID]**\*, Paula Do, Bradley Hernlem**

Western Regional Research Center, Foodborne Toxin Detection & Prevention Research Unit, Agricultural Research Service, United States Department of Agriculture, Albany, CA, United States of America

\* reuven.rasooly@ars.usda.gov

## Abstract

*Bacillus cereus* is a foodborne pathogen causing emesis and diarrhea in those affected. It is assumed that the non-hemolytic enterotoxin (Nhe) plays a key role in *B. cereus* induced diarrhea. The ability to trace Nhe activity is important for food safety. While assays such as PCR and ELISA exist to detect Nhe, those methods cannot differentiate between active and inactive forms of Nhe. The existing rabbit ileal loop bioassay used to detect Nhe activity is ethically disfavored because it uses live experimental animals. Here we present a custom built low-cost CCD based luminometer and applied it in conjunction with a cell-based assay using Vero cells transduced to express the luciferase enzyme. The activity of Nhe was measured as its ability to inhibit synthesis of luciferase as quantified by reduction of light emission by the luciferase reaction. Emitted light intensity was observed to be inversely proportional to Nhe concentration over a range of 7 ng/ml to 125 ng/ml, with a limit of detection of 7 ng/ml Nhe.

## Introduction

*Bacillus cereus* is a toxin-producing Gram-positive, spore-forming, motile, aerobic rod that also grows well anaerobically. Spores of *B. cereus* are present in the soil at high concentrations up to $10^6$ cfu g$^{-1}$ [1] and enter the human food chain by contaminating crops used for feed and food. Foods observed to be affected by this contamination include the following: cooked rice [2], infant rice cereal [3], infant formula [4], dried milk products [5], dehydrated potato products [6], eggs, meat and spices [7], causing 47% of the total cases of food poisoning in Iceland (1985–1992), 22% in Finland (1992), 8.5% in the Netherlands (1991), and 5% in Denmark (1990–92) [8]. Most reported outbreaks involving *B. cereus* were linked to the consumption of heat-treated foods [9] (EFSA 2016) and between 400 and $10^8$ *B. cereus* CFU/g were found in the foods found responsible in those foodborne disease outbreaks [10]. Thomas et al. estimate that the true food poisoning incidence caused by *B. cereus* is underdiagnosed and is likely to be under-reported [11]. The Centers for Disease Control (CDC) provides estimates that there are over 63,400 cases annually within the United States of food poisoning caused by eating *B. cereus* contaminated food [12]. Daelman et al., show that the proper food handling is adequate

**Funding:** This project was supported by the United States Department of Agriculture, Agricultural

Research Service (ARS). The authors are all employed by the ARS. There was no additional source of external funding. The funders had no role in study design, data collection and analysis, decision to publish, or preparation of the manuscript.

**Competing interests:** The authors have declared that no competing interests exist.

to prevent illnesses caused by *B. cereus* [13]. Samapundo et al., showed that 88% of the isolates they studied did not grow at ≤8˚C [14] and Guérin A et al., showed that toxin produced by *B. cereus* increased 5-fold between 8˚C and 10–15˚C and by more than 100-fold between 15˚C and 25˚C while production of toxins is not favorable under anaerobic conditions [15]. Guérin A et al., 2016 shows that combinations of anaerobiosis with low pH and cold temperatures effect the growth capacities of *Bacillus cereus* [16]. Toxin produced by *B. cereus* has been linked to acute liver failures and acute encephalopathy [17–21].

This bacterium produces a group of virulence factors causing two different types of gastrointestinal illness, manifested either by emesis or diarrhea [22–24]. Emesis is usually associated with cereulide, a cyclic dodecadepsipeptide, and diarrheal symptoms associated with the enterotoxins hemolysin BL (HBL) and non-hemolytic enterotoxin (Nhe) [25–27]. In a study of 100 *B. cereus* isolates, Moravek et al., found that nearly all produced Nhe and that there was little statistical difference in the toxic activity of isolates expressing HBL and Nhe from those expressing just Nhe [28]. It is assumed that the non-hemolytic enterotoxin (Nhe), the major and the first enterotoxin secreted during the exponential growth phase by this bacterium [29], plays a key role in *B. cereus* induced diarrhea [30]. Nhe is comprised of three protein components: NheA (41.0 kDa), NheB (39.8 kDa), and NheC (36.5 kDa). Both NheB and NheC have been identified as cell binding components. After NheC and NheB bind to the cell surface they undergo a conformational change, altering the shape of their protein structure, allowing subsequent binding of the third component NheA, resulting in cell lysis [31].

The existing assay to detect the diarrheal activity of *B. cereus* virulence factors is an *in vivo* rabbit ileal loop bioassay. This process includes ligating a segment of ileum into which is transferred supernatant from a culture suspect, leading to secretion of fluids into the intestinal loop [32]. Mice and rats have also been used in place of rabbits although requiring more animals for a given number of samples [27]. Because of ethical and policy concerns about the use of live experimental animals, this *in vivo* method is not preferred. An alternative method uses polymerase chain reaction (PCR) to detect the genes for the toxin but not the toxin itself. This method is rapid and has high sensitivity and specificity [33, 34] but is not quantitative. For the assessment of food safety it is important not only to monitor for the presence of the organism as through PCR but also the actual toxic potential of those foodborne strains [35]. Thus toxin quantification is necessary because there is a very wide variation in the amount of enterotoxins produced by different strains of the bacterium and PCR tests that verify the presence of genes for the toxins cannot predict the expression level of the toxin [28] nor whether those genes are associated with viable bacteria. An actual quantitative test for the toxin is best able to assess the potential toxicity of isolates. Another approach is the use of immunoassay. However the available immunoassay kits target only one of the three elements of the Nhe toxin, specifically NheA [22, 36], while NheB and NheC are necessary for the complete toxin to bind the cell membrane of its target and only with all three elements present does the toxin have activity. Because of this and also because immunoassays respond to the presence of a particular epitope and not the active conformation of that epitope, they are unable to discern active Nhe from inactive forms and are not acceptable alternatives to the ileal loop assay to demonstrate the presence of biologically active toxin [26]. An alternative *ex-vivo* bioassay method that directly measures toxin activity quantifies the uptake and incorporation of C14 radiolabeled leucine across the plasma membrane of Vero epithelial cells [24]. Toxin activity is measured by reduction of incorporation of radioactive C14 into newly synthesized protein. While this method directly detects Nhe toxin activity without the use of living animals it still requires radioisotopes and expensive equipment such as a scintillation counter to detect the uptake of C14-leucine. As an improvement to this cell based approach, in this study we generated recombinant adenoviral vectors for the expression of the firefly luciferase enzyme (Ad-Luc) and used these

to transduce Vero cells to create a light reporting system to measure Nhe inhibition of protein synthesis. We also evaluated the assay in a custom built low-cost CCD based luminometer and sensor, quantitatively measuring the emitted light intensity of the reaction catalyzed by the expression of luciferase in transduced Vero cells.

## Materials and methods

### Materials

Custom made analysis plates were fabricated from black 1/8" thick polymethyl methacrylate (PMMA), 3M™ adhesive transfer tape (double sided, #9770) and thin polycarbonate sheet, all purchased from Piedmont Plastics, Inc. (Beltsville, MD, USA). Luciferase Assay System reagent was purchased from Promega (Sunnyvale, CA, USA). *Bacillus cereus* toxin was a gift from Toxin Technology (Sarasota, FL, USA). HEK293 cells (ATCC CRL-1573), a cell line originating from human embryonic kidney, and epithelial Vero cells (ATCC CCL-81), a cell line originating from the kidney of African green monkey, were obtained from the American Type Culture Collection (Manassas, VA, USA).

### Cell culture medium

Dulbecco's Modified Eagle Medium (DMEM) Life Technologies (Grand Island, NY, USA) supplemented with 10% fetal bovine serum (FBS) and 100 units/mL penicillin and streptomycin was used for the maintenance of the Vero and HEK293 cell lines.

### Photodetector system

A simple and inexpensive device for simultaneous photodetection from an entire assay plate was constructed from a cooled astronomical CCD camera with 16-bit greyscale resolution and equipped with a 12 mm f1.2 lens. The model SXVF-M7 CCD camera was obtained from Adirondack Video Astronomy (Hudson Falls, NY, USA) and the lens was purchased from Spytown (Utopia, NY, USA). Custom assay plates were fabricated by laser machining sheets of black PMMA previously coated on one side with double sided adhesive tape (3M™ 9770). Microplate wells were laser cut as round holes through both the PMMA and tape layers. The plates were completed by applying thin polycarbonate sheet material on the surface coated with adhesive tape. To prevent or minimize the transfer of light from one well to another the PMMA sheet was selected to be black and opaque.

### Processing of images

The imaging of the low cost luminometer is achieved through the cooled CCD SXVF-M7 camera which efficiently converts photons of light into a corresponding electrical current with linear response and generates high-quality, low noise images. Processing of the images and quantification of light intensity was performed with the ImageJ software application [37]. The background signal of each pixel in the digital image was measured and averaged. Images of sample wells were similarly processed and the background signal subtracted. The background signal was recorded by capturing images with exposure and gain settings identical to those used to image samples. Each data point was computed as the mean pixel intensity of three samples.

### Adenoviral vector construction for expression of the firefly luciferase gene

To quantify Nhe activity we measured its inhibition of firefly luciferase gene expression levels in transduced Vero cells. The firefly luciferase gene was isolated from pGL3-Basic Vector

(Promega) and was subcloned into the adenoviral shuttle plasmid flanked by the adenovirus E1 sequences. This adenoviral shuttle plasmid and the plasmid pJM17 containing the full length of the adenovirus genome were co-transfected in HEK293 cells. After 10 days, structural changes in transfected HEK293 cells emerged. The cells became round and detached from the 6-well plate. The HEK293 cells with cytopathic effect were analyzed for luciferase expression.

## Quantifying Nhe activity

Aliquots of $1x10^4$ Vero cells in 100μl of medium were transferred per well onto 96-well plates or custom assay plates and incubated overnight at 37˚C in an atmosphere of 5% $CO_2$ to promote cell attachment. The attached cells were then transduced with Ad-Luc at MOI of 100 and after 1 h, Nhe was added to each well and incubated for 72 h at 37˚C in a 5% $CO_2$ incubator. The medium was removed, and cells were washed three times with pH 7.4 phosphate buffered saline (PBS). The luciferase enzyme activity was determined according the manufacturer's instructions for Luciferase Assay System (Promega, Madison, WI, USA) using a Synergy HT Multi-Detection Microplate Reader (BioTek, Winooski, VT, USA) or the CCD photodetector system.

## Statistical analysis

All experiments were repeated at least three times and one-way analysis of variance (ANOVA) was performed using SigmaStat 3.5 for Windows from Systat Software (San Jose, CA, USA) to compare between the different Nhe treatments. Statistical significance of results was established with $p < 0.05$.

## Results

The suitability of the bioluminescence assay for quantitative measurement of *B. cereus* nonhemolytic enterotoxin complex (Nhe) was first tested using a commercial microplate reader as luminometer. We transduced Vero cells with adenovirus to express the firefly luciferase gene as a reporter of protein synthesis and inhibition. The transduced Vero cells were immobilized in a 96-well plate and incubated in the presence of different Nhe concentrations. The luminescence from the cells was measured by a commercial luminometer microplate reader containing a photomultiplier tube that generates an amplified electric signal derived from photons of light striking its photosensitive cathode surface. The data presented in Fig 1, show that the light intensity (relative luminescence units) is inversely proportional to different Nhe concentrations. A dose dependent response is observed between 7 ng/ml and >62.5 ng/ml of Nhe. At Nhe concentrations higher than 125 ng/ml the toxin blocks luciferase enzyme synthesis and, therefore, the transduced Vero cells produce no light and the signal is at background level of untransduced cells. The statistical comparison test shows that the detection limit of 7 ng/mL has P-value < 0.05 compared to transduced cells without Nhe present.

We next examined the suitability of the low-cost CCD based luminometer as an alternative to the photomultiplier based commercial microplate reader for quantitative measurement of the light intensity from transduced Vero cells. The identical procedure was repeated over the same Nhe concentration range with the assay sample mixtures being transferred to custom 9-well sample assay plates. Initially, an CCD astronomy camera from Point Grey Research was considered for quantification of the intensity of light produced by the luciferase-catalyzed luciferin reaction. These tests were disappointing as it was determined that this camera was too noisy to discern the luminescence signal levels from the background. Even with the shutter left open to obtain longer light exposure does not overcome the problem because both signal and thermal noise inherently generated in the silicon chip of the CCD increase with exposure time. We next selected an astronomical SXVF-M7 camera with a cooled CCD and producing a

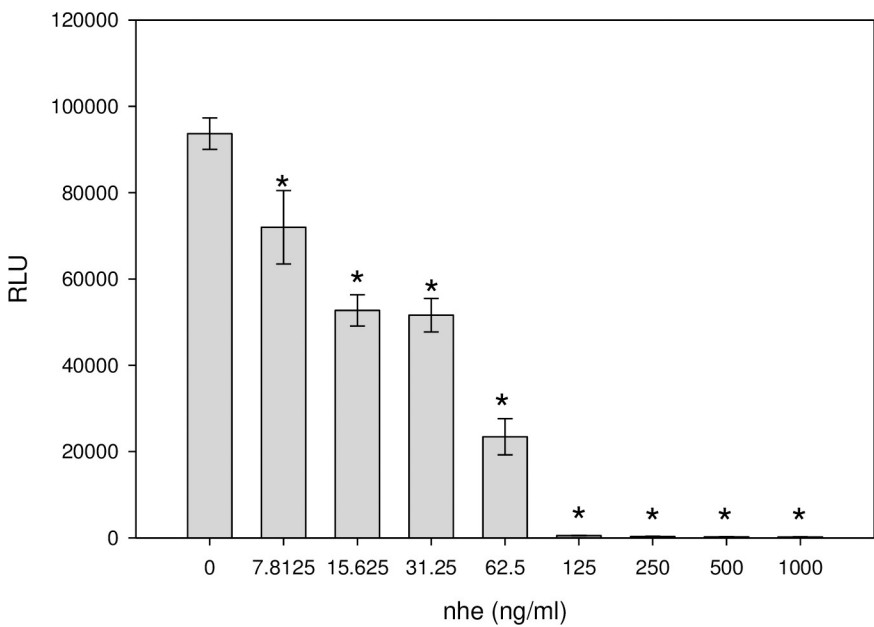

**Fig 1. Determination of the biological activity of Nhe by measuring the decrease in light from the luciferase reaction as a result of inhibition of luciferase enzyme synthesis.** Transduced Vero cells with adenoviral vectors that express the luciferase gene (Ad-Luc) were incubated with various concentrations of Nhe. The photon emission was detected by Bio Tek plate reader measured as relative light units. Values significantly different (P<0.05) from control without Nhe are marked with an asterisk. Error bars represent standard errors.

16-bit resolution grayscale image. The camera was fitted with a Pentax 12 mm f1.2 lens and connected to a computer to control acquisition of images and for their subsequent analysis. A focused image of the assay plate is projected upon the CCD chip inside the camera. The silicon CCD chip divided into an array of pixels, converts incident photons into an electronic signal proportional to photon intensity of the corresponding image. Because the camera uses Peltier cooling of the CCD chip the thermal noise background of the image is reduced. Fig 2 illustrates a digital image of an example assay plate containing transduced Vero cells after Nhe at concentrations of 1000, 500, 250, 125, 62.5, 31.25, 15.625, 7.8125 and a control 0 Nhe ng/ml were added to the well numbers 1,2, 3, 4, 5, 6, 7, 8 and 9 respectively. Those experiments were done in triplicate. The signal intensity was averaged over three replicated images over all pixels as measured and reported by the freely distributed ImageJ imaging software [37]. Similarly, an average of the background signal was calculated and was subtracted from the averages from the images of the assay samples. The resultant relative signal intensity is described in relative analog-digital units (ADU) and was plotted against concentration of Nhe. The bar graph in Fig 3 shows a proportional relationship between light intensity and Nhe concentration.

In terms of sensitivity and limit of detection, this bioluminescence assay can detect 31.25 ng/mL of Nhe with a P-value of 0.05 (p < 0.05) found by a statistical comparison test against the control case in the absence of toxin. This data compares with that for the plate reader results shown in Fig 1, although that instrument had a lower detection limit of 7.85 ng/ml. The dynamic range of the plate reader was 7.85–62.5 ng/ml compared to 31.25–125 ng/ml for the CCD system.

## Discussion

This study has been directed toward developing inexpensive methods for detecting and quantifying active *B. cereus* virulence factors that do not require the use of an *in vivo* rabbit ileal loop bioassay to avoid ethical concerns regarding the use of experimental animals. Our previous

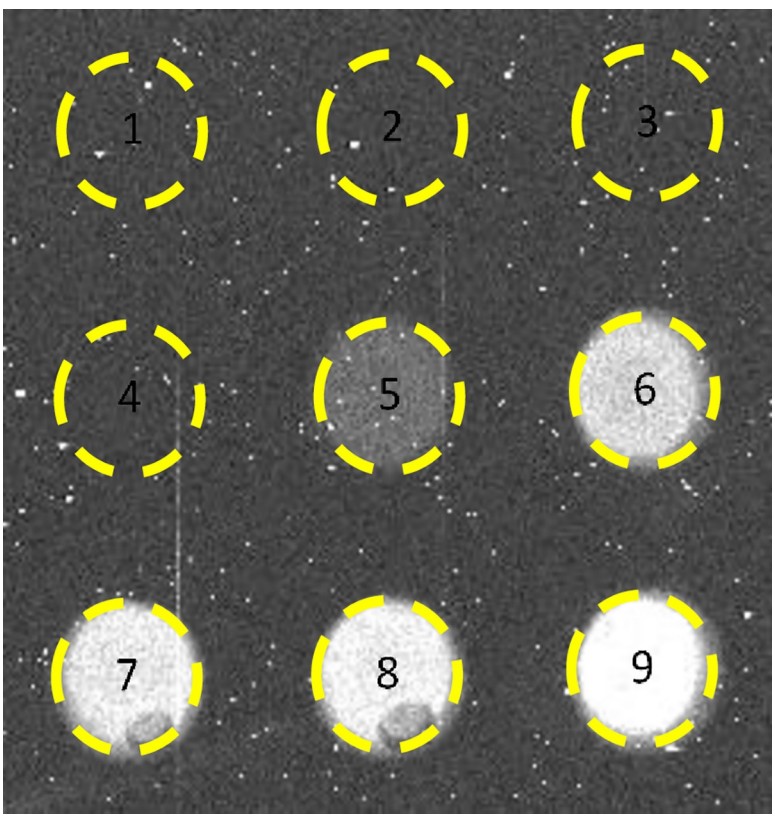

**Fig 2. Image of an example custom bioluminescence assay plate that shows an inverse dose-response relationship between Nhe concentration (wells 1–9 highest to lowest) and increasing levels of bioluminescence intensity from the reaction mixture in the custom assay plate wells.**

work has focused on the fact that *B. cereus* secretes virulence factors that inhibit protein synthesis when grown at 37˚C for 24 h in Luria Broth nutritionally rich medium [38]. The present study is focused on the virulence factor of purified Nhe. We generated adenoviral vectors that encode and express firefly luciferase enzyme (Ad-Luc) that can be used to transduce the Vero cell line to create a light reporting system to measure Nhe inhibition of protein synthesis. We also constructed a low-cost CCD based luminometer and sensor to quantitatively measure the luciferase light emission intensity from transduced Vero cells. The data presented here show that purified Nhe at a concentration of 62.5 ng/ml has the same protein synthesis inhibition effect as the secreted *B. cereus* virulence factors diluted 16 times [38]. That fact suggests that the concentration of virulence factors secreted after 24 h by *B. cereus* have the effective activity of 1000 ng/ml purified Nhe.

Alternative detection methods for Nhe include ELISA, PCR and lateral flow apparatus (LFA). ELISAs have a comparable detection limit to our assay with a range of 2–5 ng/ml and the LFA has a detection limit around 20 ng/ml [35]. It is difficult to determine a detection limit for PCR, but has been shown to produce a negative result in samples where the LFA will produce a positive result. This indicates that the detection limit of PCR is higher than that of 20 ng/ml. However, none of these methods demonstrate the presence of the biologically active form of Nhe and PCR and LFA produce only qualitative results. Our assay involves a 72 hour incubation. A shorter incubation could be used but would reduce the sensitivity of the assay.

The data presented here show that a low cost cooled CCD camera designed for astrophotography utilized in combination with this cell based assay expressing firefly luciferase enzyme

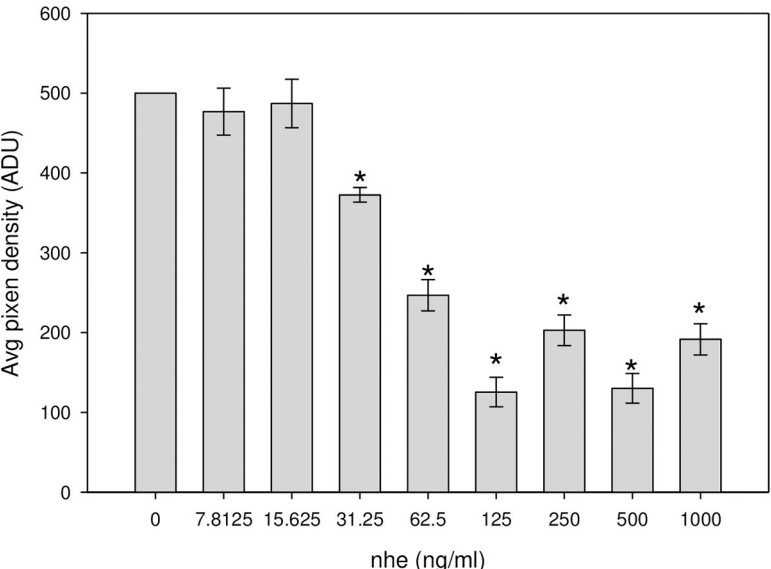

**Fig 3. Biological activity of Nhe determined by measurement of the decrease in luciferase reaction generated light resulting from inhibition of protein synthesis.** Transduced Vero cells with Ad-Luc were incubated with various concentrations of Nhe. Bioluminescence was detected using an SXVF-M7 CCD camera and expressed in average pixel density units. Values significantly different (P<0.05) from control without Nhe are marked with an asterisk. Error bars represent standard errors.

can be used for quantitative measurement of the bioluminescence response to Nhe activity at levels similar to sensitive commercial luminometers costing 30 times more than a CCD device. Adoption of this technique is especially suitable and appropriate in locations where resources are limited. Additionally, the simplicity and reduced cost facilitates expanded Nhe testing for the advancement of food safety. However, this assay will need a cell culture facility and a freezer for the substrate. A further advantage of the CCD device over the luminometer is in the effective multiplexing of data measurements; the CCD device can measure the light emission of multiple tests at the same time whereas the luminometer is a sequential analytic machine.

The light levels generated by the luciferase catalyzed oxidation of luciferin to oxyluciferin were found to be inversely correlated to Nhe concentration with a linear correlation of $R^2 = 0.99$. Biologically active Nhe was assayed with a detection limit of 7 ng/mL using this method and cell-based assay. This approach has a distinct advantage over the qualitative *in vivo* rabbit ileal loop bioassay based on the observation of diarrhea produced in response to administered *B. cereus* secreted virulence factors that disrupt the integrity of the plasma membrane of epithelial cells in the small intestine.

## Supporting information

**S1 Table.**
(XLS)

**S2 Table.**
(XLS)

## Acknowledgments

We thank Daphne Tamar and Sharon Abigail for their inspiration.

## Author Contributions

**Investigation:** Reuven Rasooly.

**Methodology:** Paula Do, Bradley Hernlem.

**Writing – original draft:** Reuven Rasooly.

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
