## [Decision Letter · Decision Letter 0]

23 Jul 2020

PONE-D-20-15755

Quantitative Bioluminescence Assay for Measuring Bacillus cereus Nonhemolytic Enterotoxin Complex

PLOS ONE

Dear Dr. Reuven Rasooly,

Thank you for submitting your manuscript to PLOS ONE. After careful consideration, we feel that it has merit but does not fully meet PLOS ONE’s publication criteria as it currently stands. Therefore, we invite you to submit a revised version of the manuscript mainly addressing the following points raised during the revision:

1. Update the information related to the impact of B. cereus as a food spoilage in human health.

2. Comment on how the new method compares with PCR/ELISA for cases with B. cereus presence and different toxin levels.

3. Comment on possible alternatives (kinetics, assay set-up, etc) 

We look forward to receiving your revised manuscript.

Kind regards,

Maria Gasset, Ph.D.

Academic Editor

PLOS ONE

Journal Requirements:

Reviewers' comments:

Reviewer's Responses to Questions

**Comments to the Author**

1. Is the manuscript technically sound, and do the data support the conclusions?

Reviewer #1: Yes

Reviewer #2: Yes

2. Has the statistical analysis been performed appropriately and rigorously? 

Reviewer #1: Yes

Reviewer #2: Yes

3. Have the authors made all data underlying the findings in their manuscript fully available?

Reviewer #1: Yes

Reviewer #2: Yes

4. Is the manuscript presented in an intelligible fashion and written in standard English?

Reviewer #1: Yes

Reviewer #2: Yes

5. Review Comments to the Author

Reviewer #1: It is a very well written paper describing new approaches to detect and quantify the non-hemolytic enterotoxin of Bacillus cereus, a foodborne and highly dangerous toxin. The manuscript describes the design of not only a new assay, based on th inhibition of luciferase synthesis, but also describes the construction of a detection devise. I have no further comments to do and, in my opinion, the manuscript has enough merit and quality as to deserve publication as it is.

Reviewer #2: The authors proposed a new strategy to detect active B.cereus toxin by inhibition of the luciferase synthesis in mammalian cells transfected with its gene. Similar techniques have been already used in vitro for the detection and evaluation of other membrane-active substances, including toxins. Also, a homemade cheap luminometer to carry out the evaluation of the active toxin in situ was described.

The article is well written, with a right statistical approach, and original. Nevertheless the authors must answer important questions before approval for publication

1.- The scope of the article fits much better the scope of a Food technology journal rather than PLOS One, although it is a full decision from the Editor .

2.- The information provided in the Introduction about the impact of B.cereus as a food spoilage in human heath are outdated, the authors are advised to provide more recent information.

2,.- No doubt, the aim of the authors is to remark the practical implementation of their technique, and the advance provided by their approach. In order to do that, the drawbacks for the other techniques, animal model, PCR and ELISA were described, but they did not compared the sensitivity of at least PCR and ELISA with their method. From my point of view, just the presence of Bacillus cereus regardless of the level of toxin production is by itself a serious threat in food contamination. Does the new procedure challenge these two other techniques in terms of sensitivity?

3.- According to the protocol proposed transfection plus 72 h incubation with the toxin is required. Is this time frame affordable for groceries compared with ELISA or PCR?. Have the authors tried shorter time frames?.

4.- The authors described the problems associated to the practical use of ELISA or PCR in terms of the facilities required, but for this method a cell culture facility is needed , and also the t instability of the reagent solution for luciferase, especially for ATP, its substrate, that required to be frozen till its immediate use is obviated. In order to provide a fair scenario for this technique, these drawbacks must to be included, and if possible, their feasible solutions commented in the text.

6. PLOS authors have the option to publish the peer review history of their article (what does this mean?). If published, this will include your full peer review and any attached files.

Reviewer #1: **Yes: **Álvaro Martínez-del-Pozo

Reviewer #2: No

---

## [Author Response · Author response to Decision Letter 0]

5 Aug 2020

Dear Academic Editor Dr. Maria Gasset,

Thank you for considering our manuscript entitled: “Quantitative Bioluminescence Assay for Measuring Bacillus cereus Nonhemolytic Enterotoxin Complex” (ID: PONE-D-20-15755) for publication as a research article in PLOS ONE. We have addressed the questions posed by the Reviewers and would like to thank Reviewers for their constructive comments. We feel that the manuscript has greatly benefited as a result. 

Reviewer #1 raised no objections to publication as is. Reviewer #2 raised the following points and we have addressed them as noted.

… the authors must answer important questions before approval for publication

1.- The scope of the article fits much better the scope of a Food technology journal rather than PLOS One, although it is a full decision from the Editor.

The authors recognize the broad scope of the journal PLOS ONE and the approach described may have broader analytic potential application. While the bioluminescence assay was demonstrated in the manuscript for measuring nonhemolytic enterotoxin activity, similar cell-based assays combined with low-cost optical sensing device can be adapted for the analysis of activity of other biomolecules and different type of toxins as an alternative to live animal testing. We added this important point to the discussion. We believe the broad potential application of the approach will be of interest to the journal’s readers.

2.- The information provided in the Introduction about the impact of B. cereus as a food spoilage in human heath are outdated, the authors are advised to provide more 

We have substantially expanded the introduction, specifically including more discussion of the impact of B. cereus and its toxins on human health and the need for improved alternatives to live animal testing for assessment of the toxic potential of isolates and contaminated foodstuffs. We have expanded the reference set accordingly with recent literature on the subject. Food is routinely heat treated to kill contaminant microbial pathogens and to inactivate their toxins. The main drawback of PCR and ELISA assays is their inability to differentiate between active toxin from inactivated toxin. PCR assay can detect the presence of the genetic material of non-viable microbial pathogens encoding toxin long after heat treatment already killed them. ELISA assay detects the presence of one of the three toxin subunits which poses no threat by itself. The activity bioassay that we developed distinguishes between the biological active form of the toxin, which poses a threat, from the inactive form, which poses no threat. 

2,.- No doubt, the aim of the authors is to remark the practical implementation of their technique, and the advance provided by their approach. In order to do that, the drawbacks for the other techniques, animal model, PCR and ELISA were described, but they did not compared the sensitivity of at least PCR and ELISA with their method. From my point of view, just the presence of Bacillus cereus regardless of the level of toxin production is by itself a serious threat in food contamination. Does the new procedure challenge these two other techniques in terms of sensitivity?

In terms of limit of detection, our bioluminescence assay can detect 7 ng/ml Nhe. This data compares with ELISA, PCR and lateral flow apparatus (LFA). ELISAs have a comparable detection limit to our assay with a range of 2-5 ng/ml and the LFA has a detection limit around 20 ng/ml [35]. It is difficult to determine a detection limit for PCR, but has been shown to produce a negative result in samples where the LFA will produce a positive result. This indicates that the detection limit of PCR is higher than that of 20 ng/ml. However, none of these methods demonstrate the presence of the biologically active form of Nhe and PCR and LFA produce only qualitative results. Further, commercially available tests only show the presence of one sub-unit of the Nhe complex. We have included further clarification of these points in the Introduction and Discussion.

3.- According to the protocol proposed transfection plus 72 h incubation with the toxin is required. Is this time frame affordable for groceries compared with ELISA or PCR?. Have the authors tried shorter time frames?.

A shorter time frame will reduce the sensitivity of the assay. For the application of measuring toxin activity in food, the speed may not be so relevant because at the time of the measurement the presence of the toxin can already be established by immunologic screening assays, the relevant question remaining is whether the detected toxin is active and presents a health risk or whether the toxin is inactive.

4.- The authors described the problems associated to the practical use of ELISA or PCR in terms of the facilities required, but for this method a cell culture facility is needed , and also the t instability of the reagent solution for luciferase, especially for ATP, its substrate, that required to be frozen till its immediate use is obviated. In order to provide a fair scenario for this technique, these drawbacks must to be included, and if possible, their feasible solutions commented in the text.

We have pointed out these additional requirement in the Discussion. However, it is worth stressing again that the method presented is more accurately described as an alternative to live animal testing for fully functional and active toxin and not as an alternative to PCR and ELISA which do not serve that purpose.

We thank the reviewers for their constructive comments.

Sincerely, 

Reuven Rasooly, Ph. D.

---

## [Editor Report · Decision Letter 1]

11 Aug 2020

Quantitative Bioluminescence Assay for Measuring Bacillus cereus Nonhemolytic Enterotoxin Complex

PONE-D-20-15755R1

Dear Dr. Reuven Rasooly,

We’re pleased to inform you that your manuscript has been judged scientifically suitable for publication and will be formally accepted for publication once it meets all outstanding technical requirements.

Kind regards,

Maria Gasset, Ph.D.

Academic Editor

PLOS ONE
---

## [Editor Report · Acceptance letter]

15 Sep 2020

PONE-D-20-15755R1 

Quantitative Bioluminescence Assay for Measuring *Bacillus cereus* Nonhemolytic Enterotoxin Complex

Dear Dr. Rasooly:

I'm pleased to inform you that your manuscript has been deemed suitable for publication in PLOS ONE. Congratulations! Your manuscript is now with our production department. 

Kind regards, 

on behalf of

Dr. Maria Gasset 

Academic Editor

PLOS ONE